# Comprehensive Evaluation of University Safety Culture Construction Level Based on “2–4” Model

**DOI:** 10.3390/ijerph192316145

**Published:** 2022-12-02

**Authors:** Ruili Hu, Ye Zhang, Beifang Gu

**Affiliations:** 1Department of Security, University of International Business and Economics, Beijing 100029, China; 2School of Foreign Studies, University of International Business and Economics, Beijing 100029, China; 3School of Environmental and Municipal Engineering, North China University of Water Resources and Electric Power, Zhengzhou 450046, China

**Keywords:** safety culture, “2–4” model, evaluation index system, analytic hierarchy process, fuzzy comprehensive evaluation

## Abstract

Based on the “2–4” model of accident causation, a comprehensive index system of the safety culture construction level in colleges and universities is set up. This system consists of 4 primary indicators and 28 secondary indicators. Taking a university as an example, applying the analytic hierarchy process and fuzzy comprehensive evaluation method, this study establishes a comprehensive evaluation model to evaluate the construction level of the university’s safety culture. The results show that the construction level of the university’s safety culture is consistent with the actual situation. This study provides useful insights and feasible paths for promoting the safety and stability of colleges and universities and building a higher level of safety on campus.

## 1. Research on Evaluating the Level of Safety Culture Construction in Colleges and Universities

At present, the security and stability of colleges and universities are facing unprecedented challenges. Appalling accidents such as network fraud, poisoning, suicide, and murder have occurred frequently on college campuses, sounding an alarm about the security situation in educational institutions [1,2]. From the “2–4” model of accident causation proposed by Fu Gui [3], we can see that the root cause of accidents is the lack of safety culture. Several domestic scholars have conducted relevant theoretical studies on the construction of safety culture in colleges and universities from different perspectives, but no systematic index evaluation system has been constructed. The lack of such a system hinders the ability of colleges and universities to evaluate their current construction level of safety culture. Therefore, constructing a reasonable index system applicable to the level of safety culture construction in these institutions is significant to ensure safe and harmonious campuses.

Thus far, most scholars have used qualitative methods to evaluate the safety culture, and there is a lack of attention to the influence relationship between indicators, resulting in poor objectivity of the results. As quantitative research methods, the analytic hierarchy process (AHP) and fuzzy comprehensive evaluation method are mainly used to study marine transportation, electric power, and other enterprises. To comprehensively and systematically evaluate the construction level of the security culture in colleges and universities, we applied the AHP and fuzzy comprehensive evaluation method [4] to construct the evaluation model of the security culture in this university, and the affiliation theory of fuzzy mathematics was used to transform the qualitative study of the security culture into a quantitative assessment. This study has practical significance for improving the safety and security work in colleges and universities.

### 1.1. Definition of Safety Culture in Colleges and Universities

Since 1986, when the concept of “safety culture” was first proposed by INSAG [5], scholars have studied it from different perspectives. In a broad sense, safety culture includes “internalized” humanistic qualities, such as safety philosophy, safety awareness, safety values, and “externalized” humanistic expressions and carriers, such as safety management systems, behavior culture, behavior habits, standards, and others [6,7]. Safety culture, in a narrow sense, refers to the comprehensive product of individual and collective values, attitudes, abilities, and behaviors, which mainly consists of a deep-level safety concept culture, middle-level safety system culture, and surface-level safety behavior culture and safety material culture [8,9]. Safety culture, in general, informs the safety culture in colleges and universities and can be a guiding principle to promote the construction of safety and stability in campuses. By reviewing the literature, we find that most scholars conduct research based on safety culture in a broad sense and propose a safety culture construction program with safety management, safety education, or material construction as the bias. Although this construction program has a certain role in promoting the development of the safety culture in universities, problems occur such as lack of a theoretical basis, a weak sense of boundary, the duplication of construction contents, and the absence of a strict hierarchical relationship of construction contents [7].

To sum up, safety culture in colleges and universities is the sum of the safety values, system, behavior, and safety objects that are gradually formed and inherited and constantly innovated by teachers and students in scientific research, teaching, and education activities, with the purpose of ensuring the physical and mental health of teachers and students. Safety culture consists of ideological, institutional, behavioral, material culture, and other aspects.

### 1.2. “2–4” Model of Accident Causation

Since the development of accident-causing models, more than 10 accident-causing models have been commonly used, such as the domino and trajectory crossover models. The findings of Professor Fu Gui showed that the “2–4” model can describe the occurrence of accidents in a network system and carry out a large number of accident statistical analysis [10].

The “2–4” model of accident causation is proposed by Professor Fu Gui on the basis of the domino and Swiss cheese models, which show good application effects in tracing the cause of accidents and developing countermeasures for accident prevention [11,12,13]. The “2–4” model of accident causation suggests that the causes of accidents can be divided into two levels and four stages. The “2” refers to the causes of accidents as organizational behavior and individual member behavior, and the “4” refers to the safety culture (root cause), safety management system (underlying cause), habitual behavior (indirect cause), and one-time behavior (direct cause). These four stages constitute the behavioral chain of accident causation, as shown in Table 1.

By applying the “2–4” model, we can see that the root cause of accidents is the lack of safety culture, i.e., the deep safety concept culture plays a decisive role in the occurrence of accidents and is the guiding ideology for creating a safe campus. The safety management system, habitual behaviors, and one-time behaviors are all representations of safety culture. In other words, the deeper the teachers and students know the safety concept of the university, the better the implementation of the safety management system will be. Unsafe human behavior and the unsafe conditions of objects show a trend of reduction, thus reducing the occurrence of accidents and helping create a good situation of safety and stability.

### 1.3. Comprehensive Evaluation of the Level of Safety Culture Construction

By evaluating the effectiveness of the safety culture construction in colleges and universities, we can solve problems in the construction of safety culture. The ultimate goal is to achieve safety for all. To measure the level of college safety culture construction, we should start from the concept of college safety culture and consider its aspects such as the system, behavior, and material cultures based on the “2–4” model of accident causation. In addition, we need to reflect on the characteristics of the safety culture construction and pay attention to the situation of the safety characteristic culture construction of each unit within each university. Therefore, the construction effect of safety culture can be reflected from six aspects: the safety concept culture, safety system construction, safety behavior construction, safety material construction, safety characteristic culture construction, and safety performance base level of universities.

Accordingly, when evaluating the level of the safety culture construction in colleges and universities, only by thoroughly measuring the safety culture construction itself and its effect, and covering all of these six aspects, can the true level of the safety culture construction be derived comprehensively and accurately.

## 2. Construction of Evaluation Index System for the Level of Safety Culture Construction in Colleges and Universities

Through the literature, we found that the safety culture evaluation index system with coal mine, petroleum, and electric power enterprises as the main research objects is more complete [14,15,16]. Colleges and universities cannot copy the theory of enterprise safety construction but need to create a theoretical system of safety culture construction using their own special characteristics as determined by the special campus environment, organization system, and social status of the colleges and universities.

Based on the definition of safety culture discussed, combined with the “2–4” model of accident causation, the evaluation index system A of the safety culture construction level in universities is constructed (the whole index system is called “A”, and the following level index is called “B”). The evaluation index system A includes six aspects: the safety concept culture, safety system culture, safety behavior culture, safety material culture, safety characteristic culture, and safety performance level.

To reflect the secondary indexes of safety culture in the most comprehensive and effective way, we have reviewed numerous papers on researching the index system of safety culture, combined with relevant laws and regulations, and the special characteristics of universities themselves. We have also combined statistics on safety culture factors to construct the secondary indexes of the evaluation of safety culture construction in universities [17].

In summary, the evaluation index system of the safety culture construction level of colleges and universities includes 7 primary evaluation indexes and 28 secondary evaluation indexes. The evaluation index system of the level of safety culture construction in colleges and universities is shown in Appendix A.

According to the preceding analysis, the index system of the safety culture construction level of colleges and universities, which is constructed based on the theoretical basis of the “2–4” model of accident causation, has a close logical relationship among the indexes and a clear sense of hierarchy. No problem of repetition and crossover of contents occurs, so the index system can be used to evaluate the safety culture construction level of colleges and universities in a comprehensive and systematic way.

## 3. Fuzzy Evaluation of Level of Safety Culture Construction in Colleges and Universities

Three methods are used in determining the weight of evaluation indexes: qualitative, quantitative, and a combination of both methods [18]. Qualitative evaluation focuses on the “quality” factor in the construction of safety culture and reflects the humanistic thought in the evaluation. It is a kind of developmental evaluation, but its result is easily affected by the subjective factors of the evaluation subject. Quantitative evaluation emphasizes mathematical calculation and is more objective and standardized. However, due to the limitation of quantifiable factors, fully reflecting the status of safety culture construction is a difficult task.

This study employs AHP, a multi-objective decision-making method that combines qualitative and quantitative features to hierarchize the complex college safety culture construction decision-making system and ensure that weights are determined in a relatively objective, fair, and scientific manner. Therefore, AHP is used to determine the weights of the evaluation indicators.

The fuzzy comprehensive evaluation method [19] involves applying the fuzzy transformation principle and the basic theory of fuzzy mathematics-affiliation degree or affiliation function to describe the amount of fuzzy information of the mediated transition on the basis of the comprehensive consideration of all factors related to the evaluated objects and convert the expert qualitative evaluation into a quantitative score. The advantage of this method is that it avoids the problem of difficulty in quantifying the value of the safety culture construction indexes in colleges and universities because of the difference in human subjective judgment. Furthermore, using this method to assess the level of safety culture construction in colleges and universities is both scientific and feasible.

Taking a university as an example, we used the AHP and fuzzy comprehensive evaluation method to comprehensively evaluate the current situation of the safety culture in this university.

### 3.1. Construction of Evaluation Factor Sets

According to the evaluation index system of the university safety culture construction level shown in Table 2, the evaluation factor sets of levels 1 and 2 are constructed.

The set of evaluation factors for the first-level indicators is A = {*B*_1_, *B*_2_, *B*_3_, *B*_4_, *B*_5_, *B*_6_, *B*_7_}. The set of secondary index evaluation factors is *B_i_* = {*B_i_*_1_, *B_i_*_2_, *B_i_*_3_, …, *B_ik_*}, *i* = 1, 2, 3, 4, 5, 6, 7. K is the number of secondary indicators.

### 3.2. Creating a Fuzzy Rubric Set

Combined with the actual situation of the safety culture construction in universities, this study divides the fuzzy rubric set into five levels: *V* = {*V*1, *V*2, *V*3, *V*4, *V*5}{very good, better, average, poor, very poor}.

### 3.3. Determining the Weights of Evaluation Indexes for the Level of Safety Culture Construction in Universities

Using AHP to determine the weights of evaluation indicators is usually determined according to the following steps:

(1) Determining the judgment matrix. The judgment matrix represents the relative importance of the indicators related to a certain indicator in the previous level. According to the comparison scale table determined by Saaty, experts assign weights to the indicators at each level, then compare the indicators of each indicator layer pair by pair and finally, construct a judgment matrix.

(2) Characteristic root method of hierarchical single sorting.

Hierarchical single sorting can be reduced to the problem of computing the maximum eigenroots and eigenvectors of the judgment matrix. The maximum eigenvalue is:(1)$$\lambda_{\max}=\frac{1}{4}\sum_{i=1}^{4}\frac{{\left(AW’\right)}_{i}}{w_{i}}$$

(3) Consistency test of judgment matrix

For the consistency test of the nth-order judgment matrix, the consistency index is defined as:(2)$$CI=\frac{\lambda_{\max}-n}{n-1}$$

Taking the first-level indicators as an example, we can construct the judgment matrix A of the first-level indicators according to the above steps as follows:$$A=\left[\begin{array}{ccccccc} 1 & 2 & 1/3 & 1/4 & 5 & 6 & 7\\ 1/2 & 1 & 1/3 & 1/3 & 4 & 6 & 5\\ 3 & 3 & 1 & 3 & 5 & 6 & 7\\ 4 & 3 & 1/3 & 1 & 4 & 6 & 5\\ 1/5 & 1/4 & 1/5 & 1/4 & 1 & 2 & 3\\ 1/6 & 1/6 & 1/6 & 1/6 & 1/2 & 1 & 1/3\\ 1/7 & 1/5 & 1/5 & 1/5 & 1/3 & 3 & 1 \end{array}\right]$$

Based on the preceding steps, the following can be calculated:

Eigenvector *B* = [0.1365, 0.1356, 0.3553, 0.2497, 0.0567, 0.02875, 0.03743].
$$\lambda_{\max}=7.6801$$
$$CI=\frac{\lambda_{\max}-n}{n-1}=0.11335$$

Let RI=1.32; then, CR=CIRI=0.0859<0.1.

Then, the judgment matrix of the primary indicators show satisfactory consistency.

Similarly, the *B*_1_, *B*_2_, *B*_3_, *B*_4_, *B*_5_, *B*_6_, *B*_7_ judgment matrix is obtained.
$$B_{1}=\left[\begin{array}{ccc} 1 & 1/3 & 3\\ 3 & 1 & 5\\ 1/3 & 1/5 & 1 \end{array}\right]$$
$$B_{2}=\left[\begin{array}{cccc} 1 & 1/2 & 1 & 3\\ 2 & 1 & 3 & 3\\ 1 & 1/3 & 1 & 3\\ 1/3 & 1/3 & 1/3 & 1 \end{array}\right]$$
$$B_{3}=\left[\begin{array}{ccccc} 1 & 2 & 1/3 & 4 & 5\\ 1/2 & 1 & 1/3 & 3 & 4\\ 3 & 3 & 1 & 2 & 4\\ 1/4 & 1/3 & 1/2 & 1 & 3\\ 1/5 & 1/4 & 1/4 & 1/3 & 1 \end{array}\right]$$
$$B_{4}=\left[\begin{array}{ccccccc} 1 & 3 & 4 & 5 & 6 & 8 & 7\\ 1/3 & 1 & 3 & 4 & 5 & 7 & 6\\ 1/4 & 1/3 & 1 & 3 & 4 & 6 & 5\\ 1/5 & 1/4 & 1/3 & 1 & 3 & 5 & 4\\ 1/6 & 1/5 & 1/4 & 1/3 & 1 & 4 & 3\\ 1/8 & 1/7 & 1/6 & 1/5 & 1/4 & 1 & 1/3\\ 1/7 & 1/6 & 1/5 & 1/4 & 1/3 & 3 & 1 \end{array}\right]$$
$$B_{5}=\left[\begin{array}{cccc} 1 & 1/2 & 1 & 3\\ 2 & 1 & 3 & 3\\ 1 & 1/3 & 1 & 3\\ 1/3 & 1/3 & 1/3 & 1 \end{array}\right]$$
$$B_{6}=\left[\begin{array}{ccc} 1 & 3 & 3\\ 1/3 & 1 & 2\\ 1/3 & 1/2 & 1 \end{array}\right]$$
$$B_{7}=\left[\begin{array}{cc} 1 & 3\\ 1/3 & 1 \end{array}\right]$$

According to the preceding steps, the feature vectors of the evaluation indexes of the safety culture construction level of universities can be calculated and is summarized in Table 2.

Through calculation, we can see in Table 3 that the consistency ratio of each safety culture construction level evaluation index is less than 0.1. Thus, each judgment matrix shows satisfactory consistency.

### 3.4. Judgment of Secondary Indicators and Determination of Indicator Affiliation

Suppose that x experts have judged a certain indicator, and y of them have chosen a certain rubric level. Then, the affiliation of the indicator to the set of rubrics is y/x [20,21].

Ten experts were invited to judge each secondary indicator according to the five evaluation criteria given in the rubric set. Then, the affiliation degree of the secondary indicators under each primary indicator was calculated to construct the judgment matrix, and the judgment matrix, corresponding to each primary indicator, was as follows:$$R_{1}=\left[\begin{array}{c} V1\\ V2\\ V3 \end{array}\right]\left[\begin{array}{ccccc} 0.1 & 0.4 & 0.2 & 0.2 & 0.1\\ 0.1 & 0.3 & 0.3 & 0.1 & 0.2\\ 0 & 0.2 & 0.3 & 0.4 & 0.1 \end{array}\right]$$
$$R_{2}=\left[\begin{array}{c} V4\\ V5\\ V6\\ V7 \end{array}\right]\left[\begin{array}{ccccc} 0.4 & 0.2 & 0.1 & 0.2 & 0.1\\ 0.1 & 0.2 & 0.3 & 0.3 & 0.1\\ 0.4 & 0.2 & 0.1 & 0.2 & 0.1\\ 0 & 0.1 & 0.2 & 0.3 & 0.4 \end{array}\right]$$
$$R_{3}=\left[\begin{array}{l} \begin{array}{c} V8\\ V9\\ V10\\ V11 \end{array}\\ V12 \end{array}\right]\left[\begin{array}{ccccc} 0.2 & 0.3 & 0.1 & 0.2 & 0.2\\ 0.1 & 0.2 & 0.2 & 0.3 & 0.2\\ 0.4 & 0.2 & 0.2 & 0.1 & 0.1\\ 0.4 & 0.2 & 0.1 & 0.2 & 0.1\\ 0.3 & 0.2 & 0.2 & 0.2 & 0.1 \end{array}\right]$$
$$R_{4}=\left[\begin{array}{l} \begin{array}{c} V13\\ V14\\ V15\\ V16 \end{array}\\ V17\\ V18\\ V19 \end{array}\right]\left[\begin{array}{ccccc} 0.4 & 0.2 & 0.3 & 0.1 & 0\\ 0.4 & 0.3 & 0.2 & 0.1 & 0\\ 0.2 & 0.2 & 0.2 & 0.3 & 0.1\\ 0.5 & 0.2 & 0.1 & 0.1 & 0.1\\ 0.3 & 0.2 & 0.2 & 0.2 & 0.1\\ 0.4 & 0.3 & 0.1 & 0.2 & 0\\ 0.5 & 0.1 & 0.2 & 0.2 & 0 \end{array}\right]$$
$$R_{5}=\left[\begin{array}{c} V20\\ V21\\ V22\\ V23 \end{array}\right]\left[\begin{array}{ccccc} 0.3 & 0.2 & 0.2 & 0.2 & 0.1\\ 0.2 & 0.3 & 0.2 & 0.2 & 0.1\\ 0.3 & 0.2 & 0.2 & 0.2 & 0.1\\ 0.2 & 0.2 & 0.2 & 0.3 & 0.1 \end{array}\right]$$
$$R_{6}=\left[\begin{array}{c} V24\\ V25\\ V26 \end{array}\right]\left[\begin{array}{ccccc} 0 & 0.1 & 0.2 & 0.4 & 0.3\\ 0.1 & 0.2 & 0.6 & 0.1 & 0\\ 0 & 0.1 & 0.4 & 0.3 & 0.2 \end{array}\right]$$
$$R_{7}=\left[\begin{array}{c} V27\\ V28 \end{array}\right]\left[\begin{array}{ccccc} 0.3 & 0.2 & 0.3 & 0.2 & 0\\ 0.2 & 0.4 & 0.3 & 0.1 & 0 \end{array}\right]$$

### 3.5. Fuzzy Integrated Evaluation of First-Level Indicators

Based on the single-factor evaluation matrix of the second-level indicators and their corresponding weights, the fuzzy synthesis of each level of indicators is as follows:(3)$$U_{i}=W_{i}\times R_{i}$$

*U_i_* is the fuzzy composite judgment result of the *i*th level 1 index.

*W_i_* is the vector of weights of the second-level indicators under the *i*th level indicator. 

*R_i_* is the single-factor judgment matrix of the *i*th level indicator.

After calculation, we can see that:

*U*_1_ = [0.08938, 0.31543, 0.27395, 0.72788, 0.16333]

*U*_2_ = [0.22672, 0.27856, 0.19899, 0.25435, 0.12945]

*U*_3_ = [0.41849, 0.22765, 0.16253, 0.18069, 0.06494]

*U*_4_ = [0.37681, 0.22218, 0.22548, 0.14369, 0.03184]

*U*_5_ = [0.24545, 0.24452, 0.19992, 0.20975, 0.0996]

*U*_6_ = [0.0250, 0.12521, 0.33264, 0.30853, 0.20852]

*U*_7_ = [0.275, 0.25, 0.3, 0.175, 0]

The results of the above calculations are constituted into a new fuzzy evaluation matrix as the fuzzy evaluation matrix of the first-level indicators *R*:$$R=\left[\begin{array}{ccccc} 0.08938 & 0.31543 & 0.27395 & 0.72788 & 0.16333\\ 0.22672 & 0.27856 & 0.19899 & 0.25435 & 0.12945\\ 0.41849 & 0.22765 & 0.16253 & 0.18069 & 0.06494\\ 0.37681 & 0.22218 & 0.22548 & 0.14369 & 0.03184\\ 0.24545 & 0.24452 & 0.19992 & 0.20975 & 0.09960\\ 0.02500 & 0.12521 & 0.33264 & 0.30853 & 0.20852\\ 0.27500 & 0.25000 & 0.30000 & 0.17500 & 0.00000 \end{array}\right]$$

To judge the level of the safety culture construction of this university using two-level fuzzy synthetic evaluation, we set the weight values of *B*_1_*, B*_2_*, B*_3_*, B*_4_, *B*_5_*, B*_6_, and *B*_7_ to constitute the eigenvector *B*, the rubric for the level of the safety culture construction in universities as *U*, and the comments on the level of the safety culture construction in colleges and universities as U=B•R. The calculation shows that *U* = [0.31065, 0.24401, 0.24795, 0.26124, 0.09873].

According to the principle of maximum affiliation, the evaluation level of the university’s safety culture construction is very good.

Through actual investigation, we find that the level of the safety culture of this university is typical of exemplary universities. The safety concept of the university can be fully understood by teachers and students, and it has a rich cultural carrier. The safety system is perfect, and it can organize and carry out various forms of educational activities with a high participation rate. The annual featured activities are also the learning objects of other universities, the corresponding equipment and facilities adopt relatively advanced technology, and the accident rate is low. Accordingly, the evaluation grade of the level of the safety culture construction in this university obtained by applying fuzzy comprehensive evaluation matches the actual situation.

## 4. Conclusions

(1)The concept of safety culture has two types: broad and narrow. By combining the literature for comparison and analysis, we can see that the concept of safety culture in the narrow sense is more suitable for the actual situation of China’s colleges and universities at present; that is, the safety culture of these institutions is composed of conceptual, institutional, behavioral, and material culture. Among them, concept culture is the core, system culture is the guarantee, behavior culture is the representation, and material culture is the cornerstone.(2)Based on the theoretical basis of the “2–4” model of accident causation, the evaluation index system A of the safety culture in universities is constructed. The index system consists of 7 primary indicators and 28 secondary indicators, of which the primary indicators include the safety concept culture *B*_1_, safety system culture *B*_2_, safety behavior culture *B*_3_, safety material culture *B*_4_, safety characteristic culture *B*_5_, quality of accident statistics work *B*_6_, and basic safety performance *B*_7_.(3)Taking a university as an example and using AHP and the fuzzy comprehensive evaluation method, we established a fuzzy comprehensive evaluation model of the safety culture construction level of the university. We quantitatively evaluated the construction level of the safety culture of the university with the help of the affiliation theory of fuzzy mathematics. The evaluation results matched the actual situation after calculation and analysis, with strong reliability. The model is a reasonable and feasible evaluation method that is useful for other universities to evaluate the level of safety culture construction.

## Figures and Tables

**Table 1 ijerph-19-16145-t001:** Corresponding relationship between accident causes and behaviors.

Development Level	Behavior Development Chain	Classification Cause Chain	Accident Causal Chain	Development Result
Organizational level	Guiding behavior	Root cause	Lack of safety culture	Accident
	Operating behavior	Underlying cause	Lack of safety management system	
Individual level	Habitual behavior	Indirect cause	Lack of security knowledge	
			Poor safety habits	
			Lack of security awareness	
			Poor safety physiology	
			Poor safety mentality	
	One-time behavior	Direct cause	Unsafe human behavior	
			Unsafe condition of objects	

**Table 2 ijerph-19-16145-t002:** Summary table of characteristic vectors of safety culture evaluation indicators.

Evaluation Indicators	Eigenvector
Safety concept culture *B*_1_	[*B*_11_, *B*_12_, *B*_13_] = [0.2605, 0.6333, 0.1062]
Safety system culture *B*_2_	[*B*_21_, *B*_22_, *B*_23_, *B*_24_] = [0.2371, 0.4460, 0.2182, 0.0983]
Safety behavior culture *B*_3_	[*B*_31_, *B*_32,_ *B*_33_, *B*_34_, *B*_35_] = [0.2705, 0.1810, 0.3801, 0.1102, 0.0612]
Safety material culture *B*_4_	[*B*_41_, *B*_42,_ *B*_43_, *B*_44_, *B*_45_, *B*_46_, *B*_47_] = [0.3796, 0.2373, 0.1540, 0.1002, 0.0642, 0.0246, 0.0401]
Safety features culture *B*_5_	[*B*_51_, *B*_52_, *B*_53_, *B*_54_] = [0.2371, 0.4460, 0.2182, 0.0983]
Quality of accident statistics *B*_6_	[*B*_61_, *B*_62_, *B*_63_] = [0.5890, 0.2520, 0.1591]
Basic information on safety performance *B*_7_	[*B*_71_, *B*_72_] = [0.7500, 0.2500]

**Table 3 ijerph-19-16145-t003:** Consistency test results of evaluation indicators of safety culture construction level in colleges and universities.

Parameter	*A*	*B* _1_	*B* _2_	*B* _3_	*B* _4_	*B* _5_	*B* _6_	*B* _7_
$\lambda_{\max}$	7.6801	3.0387	4.118	5.461	7.6703	4.118	3.054	2.000
*CI*	0.11335	0.0194	0.039	0.1152	0.1117	0.039	0.027	0
*RI*	1.32	0.58	0.90	1.12	1.32	0.90	0.58	0
*CR*	0.0859	0.0334	0.0433	0.1029	0.0846	0.0433	0.0466	0

## Data Availability

The data that support the findings of this study are available from the corresponding author upon reasonable request.

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
