# Peer review of "Comprehensive Evaluation of University Safety Culture Construction Level Based on “2–4” Model"

_ijerph, 2022, doi:10.3390/ijerph192316145_

Round 1

Reviewer 1 Report

Overall, it is a useful study. The model and analysis methods are detailed and reliable. The manuscript can be accepted with the following revisions.

1. There is a grammatical error in line 35, please verify.

2. The meaning of line 64 is ambiguous, please confirm.

3. Why should the evaluation index of the level of safety culture construction in colleges and universities mention the construction of safety characteristic culture?

4. There is a problem with the format of Table 2,please modify it.

5. How is the judgment matrix A of the first-level index constructed?

6. What is the basis of the average random index?

7. The conclusion should be more concise.

8. What are the advantages of this research?

9. The idea of the authors of the paper need to be involved in the research, design, or to obtain data, as well as the analysis and interpretation of data; The author needs to write papers or participate in important content changes. So please check all of the authors by the rules of the authors.

10.  Please explain the advantages of the index system of safety culture construction level of colleges and universities based on the 2-4 model of accident causation.

Author Response

Thank you for reviewing our paper. We appreciate your insightful comments and suggestions. After careful discussion of the comments, we have revised the manuscript extensively. Pondering the comments has greatly improved the manuscript. In addition, we consulted a professional editing service and asked several colleagues who are native English editors to help polish our article. We hope that the current version is acceptable for publication.

To facilitate your review, we revised the paper using the revision model. The main corrections in the paper and the responds to the reviewer’s comments are as follows:

Comment 1: There is a grammatical error in line 35, please verify.

Response 1: The statement of "but no systematic index evaluation system has been formed" has been corrected as "but no systematic index evaluation system has been constructed" and this sentence is currently on line 37 in the revised version.

Comment 2: The meaning of line 64 is ambiguous, please confirm.

Response 2: The statement of " Through combing the literature "has been changed to " By reviewing the literature " and this sentence is currently on line 75 in the revised version.

Comment 3: Why should the evaluation index of the level of safety culture construction in colleges and universities mention the construction of safety characteristic culture?

Response 3: The nature of each university is different, and the emphasis of safety culture construction is also different, that's why safety characteristic culture is involved.

Comment 4: There is a problem with the format of Table 2, please modify it.

Response 4: I checked Table 2 carefully and modified the format. Now it is placed at the end of the paper as Appendix 1.

Comment 5: How is the judgment matrix A of the first-level index constructed?

Response 5: Two indicators Xi and Xj of the same indicator level are selected, and the ratio of the size of the influence of indicators Xi and Xj on the indicator level is expressed by rij. All the comparison results of the same indicator level are expressed by a matrix, and this matrix is the judgment matrix.

Comment 6: What is the basis of the average random index?

Response 6: The average random consistency index varies with the matrix, such as a 7th order matrix with RI=1.32.

Comment 7: The conclusion should be more concise.

Response 7: The conclusion has been simplified.

Comment 8: What are the advantages of this research?

Response 8: Based on analytic hierarchy process and fuzzy comprehensive evaluation method, a comprehensive evaluation model is established for the university to further evaluate the construction level of the university's safety culture. The results show that the construction level of the school's safety culture is consistent with the actual situation and this achievement provides useful ideas and feasible paths for promoting the safety and stability of colleges and universities and building a higher level of safe campus.

Comment 9: The idea of the authors of the paper need to be involved in the research, design, or to obtain data, as well as the analysis and interpretation of data; The author needs to write papers or participate in important content changes. So please check all of the authors by the rules of the authors.

Response 9: The idea of the authors has been involved in the paper, and the information of the authors has been checked.

Comment 10: Please explain the advantages of the index system of safety culture construction level of colleges and universities based on the 2-4 model of accident causation.

Response 10: The index system of safety culture construction level of colleges and universities, which is constructed based on the theoretical basis of "2-4" model of accident causation, has a close logical relationship among the indexes and a clear sense of hierarchy, and there is no problem of repetition and crossover of contents, so it can evaluate the safety culture construction level of colleges and universities in a comprehensive and systematic way.

Thank you for your insightful comments. We hope thar our responses and edits are satisfactory.

Best wishes.

Kind regards,

Authors.

Reviewer 2 Report

Thank you for inviting me to be a reviewer of the manuscript entitled Comprehensive Evaluation of University Safety Culture Construction Level Based on "2-4" Model. This document is really impressive in terms of your efforts to demonstrate the power of your study.

I recommend placing Table 2 in an appendix to this study.

I see great potential in the study for further follow-up research.

However, some passages in the introduction of the study are very descriptive and lengthy. This is sometimes confusing. Therefore, I would suggest shortening and simplifying them. On the other hand, I propose to expand the research part of this study and focus primarily on discussions of results and findings.

The study does not include a discussion of the results and a comparison with other similar research. I recommend completing this important part of the work.

This study refers to 17 scientific references, sources and publications. The references used are up-to-date, but not of sufficient quality in the field of international research. I recommend expanding the number of references used.

The basic ideas of the submitted manuscript are interesting.

Author Response

On behalf of my co-authors, I would like to express our sincere appreciations of reviewer' constructive comments and suggestions concerning our manuscript. We have studied reviewer ' s comments carefully and have tried our best to revise our manuscript using the revision model, which we hope meet with approval. In addition, we consulted a professional editing service and asked several colleagues who are native English editors to help polish our article. The main corrections in the paper and the responds to the reviewer ' s comments are as follows:

Comment 1: I recommend placing Table 2 in an appendix to this study.

Response 1: As suggested by the reviewer, I have attached Table 2 as Appendix 1 at the end of the paper.

Comment 2: I see great potential in the study for further follow-up research. However, some passages in the introduction of the study are very descriptive and lengthy. This is sometimes confusing. Therefore, I would suggest shortening and simplifying them. On the other hand, I propose to expand the research part of this study and focus primarily on discussions of results and findings.

Response 2: According to the reviewer's suggestion, we checked the paper carefully. On the one hand, We have shortened and simplified the introduction. On the other hand, we have enriched and expanded the discussions of results and findings.

Comment 3: The study does not include a discussion of the results and a comparison with other similar research. I recommend completing this important part of the work.

Response 3: By referring to a large number of literatures, the following contents are added:

(1)By comparing the "2-4" model with other accident causative models, it is known that the "2-4" model can describe the occurrence of accidents in a systematic network system and carry out a large number of accident statistical analysis. This is the advantage of creating an evaluation index system of safety culture construction level in colleges and universities based on the "2-4" model. The created evaluation index system of safety culture construction level has the characteristics of systematization and close logical relationship between each index.

(2) The application industries of analytic hierarchy process and fuzzy comprehensive evaluation method are added. As a commonly used evaluation method, analytic hierarchy process and fuzzy comprehensive evaluation method is mainly used to evaluate the safety culture construction level of Marine transportation, electric power and other enterprises. It is an innovative attempt to use analytic hierarchy process and fuzzy comprehensive evaluation method to quantitatively evaluate the safety culture construction level of colleges and universities.

Comment 4: This study refers to 17 scientific references, sources and publications. The references used are up-to-date, but not of sufficient quality in the field of international research. I recommend expanding the number of references used.

Response 4: We have checked the literature carefully and completed the work of adding references about international research.

Once again, thank you very much for your comments and suggestions.

Best wishes.

Kind regards,

Authors.

Round 2

Reviewer 2 Report

The study was corrected and supplemented. I agree to publish.